# Dr. LLM Will See You Now: The Ability of ChatGPT to Provide Geographically Tailored Colorectal Cancer Screening and Surveillance Recommendations

**DOI:** 10.3390/jcm14145101

**Published:** 2025-07-18

**Authors:** Aisling Zeng, Jacqueline Steinke, Horea-Florin Bocse, Matteo De Pastena

**Affiliations:** 1Michael G. DeGroote School of Medicine, McMaster University, Hamilton, ON L8S 4L8, Canada; aisling.zeng@medportal.ca; 2Division of General/Colorectal Surgery, Ashford and St. Peter’s Hospitals NHS Foundation Trust, Chertsey KT16 0PZ, UK; jacqueline.steinke@nhs.net; 3Department of General Surgery, Regional Institute of Gastroenterology and Hepatology, 400162 Cluj-Napoca, Romania; horea.f.bocse@gmail.com; 4Department of General and Pancreatic Surgery, The Pancreas Institute, University of Verona Hospital Trust, 37129 Verona, Italy

**Keywords:** colorectal neoplasms, surgery, colonoscopy, large language models, guidelines, ChatGPT, artificial intelligence

## Abstract

**Background/Objectives**: This study evaluates the performance of a large language model (lLm) in providing geographically tailored colorectal cancer screening and surveillance recommendations to gastrointestinal surgeons. **Methods**: Fifty-four patient cases, varying by age and family history, were developed based on colorectal cancer guidelines. Standardized prompts with predefined query terms were used to query ChatGPT-4.5 on 18 April 2025, from four locations: Canada, Italy, Romania, and the United Kingdom. Responses were classified as “Correct,” “Partially Correct,” or “Incorrect” based on clinical guidelines and expert recommendations for each country. Outcomes were analyzed using descriptive statistics. **Results**: ChatGPT provided recommendations on screening eligibility, test interpretation, the management of positive results, and surveillance intervals. Correct recommendations were given for 50.0% (27/54) of cases in Canada, 63.0% (34/54) of cases in Italy, 40.7% (22/54) of cases in Romania, and 55.6% (30/54) of cases in the United Kingdom. Queries in Italian yielded correct guidance for 64.8% (35/54) of cases, while Romanian queries were accurate for 40.7% (22/54) of cases. Notably, Romania and Italy lacked detailed guidelines for polyp management and post-test surveillance. A key finding was the inconsistency between ChatGPT-generated titles and corresponding recommendations, which may impact its reliability in clinical decision-making. **Conclusions**: ChatGPT-4.5’s performance varies by country and language, highlighting inconsistencies in geographically tailored recommendations. This study highlights limitations associated with the training data cutoff and the potential biases introduced by model-generated responses. Healthcare professionals should recognize these limitations and the possible gaps in guideline availability, particularly for high-risk screening, polyp management, and surveillance in certain European countries.

## 1. Introduction

The accessibility and rapid response capabilities of large language models (LLMs) make them appealing as clinical decision-making tools for both physicians and patients [1,2,3,4]. However, these artificial intelligence (AI) models are primarily trained on non-medical, publicly available datasets, often with a predominant focus on English-language sources [5]. Despite their potential, concerns persist regarding their accuracy, limitations in training data, and inconsistencies across different geographical contexts [6,7,8]. Additionally, many of the most widely used chatbots operate as closed-source LLMs owned by private companies, with their characteristics and training datasets concealed from public scrutiny [5,8]. The training data may significantly influence the responses of LLMs to medical queries, potentially impacting patient decision-making and health outcomes [8]. Consequently, their ability to provide clinically accurate and geographically specific medical recommendations remains uncertain.

Previous studies have shown that LLMs provide accurate colorectal cancer screening recommendations. These observed differences in accuracy may be due to several contributing factors: regional variations in clinical guidelines, differences in the availability and dissemination of national recommendations, language-specific training limitations, and regional biases in the underlying training datasets of the LLMs with varying success rates, ranging from 22% to 77% [9]. Pereyra et al. found that ChatGPT correctly answered 45% of colorectal cancer-related questions in Argentina [10], whereas Choo et al. reported an accuracy rate of 86.7% in South Korea [11]. These variations may stem from differences in clinical practice standards, which can fluctuate between cities, regions, countries, and continents. However, it remains unclear whether these disparities are primarily due to geographical differences. ChatGPT-4.5 is capable of understanding and responding in over 50 languages, with reported accuracy rates of 85.5% in English and 84.1% in Italian, according to the Massive Multitask Language Understanding (MMLU) benchmark evaluation [5,12]. Despite this, many languages supported by ChatGPT-4.5 have not been rigorously evaluated through the MMLU, leaving gaps in knowledge regarding its clinical performance across different locations and languages [5,13].

As ChatGPT continues to be utilized worldwide, robust methodologies must be implemented to assess its clinical accuracy and reliability in different geographical and linguistic contexts. Understanding the variations in LLM-generated clinical decision-making is crucial as AI becomes increasingly integrated into medical practice. Indeed, ChatGPT could serve as a complementary tool to AI. Although it does not analyze images, it can support clinicians by interpreting clinical scenarios and applying complex, country-specific guidelines to offer personalized recommendations [14]. For example, when a lesion is detected and removed, deciding the proper surveillance interval requires considering factors such as patient age, polyp type, number, and family history—areas where ChatGPT can integrate information from clinical guidelines.

Mixed-methods research has gained traction in healthcare, enabling the integration of quantitative performance metrics with qualitative insights to provide a more comprehensive analysis [15,16]. By employing a mixed-methods approach, this study not only quantifies the accuracy of LLMs but also examines the contextual factors that influence their recommendations. Inaccurate AI-generated recommendations may lead to inappropriate screening intervals, missed diagnoses of early-stage colorectal cancer, or unnecessary procedures, all of which carry substantial clinical and economic consequences. Thus, evaluating the reliability of LLMs in this domain is both urgent and essential. Therefore, this study aimed to evaluate the performance of ChatGPT-4.5 in generating recommendations for colorectal cancer screening and surveillance across various countries and languages.

## 2. Materials and Methods

### 2.1. Study Design

This study follows a qualitative mixed-methods approach, adhering to established guidelines for assessing AI-based clinical decision support tools [17] (Appendix A). We adopted elements of the Developmental and Exploratory Clinical Investigations of Decision Support Systems Driven by Artificial Intelligence (DECIDE-AI) for standardization and reproducibility [18] (Appendix A).

### 2.2. LLM Selection and Objectives

The study team selected ChatGPT (OpenAI OpCo, LLC, 1455 3rd Street, San Francisco, CA 94158, USA) as the intervention due to its widespread accessibility and frequent use among clinicians and patients. The primary objective was to determine the accuracy of ChatGPT in providing clinicians with colorectal cancer screening and surveillance recommendations for hypothetical patient cases. This was achieved by querying ChatGPT in different countries and languages and comparing its responses against country-specific guidelines.

### 2.3. Prompt Engineering and Testing

Hypothetical patient cases were developed in accordance with established clinical practice guidelines. However, it is acknowledged that hypothetical scenarios may not capture the full complexity and contextual nuances of real-world clinical encounters, incorporating key variables such as population, intervention, and comparators [19,20,21]. The prompts were phrased to reflect standard medical terminology at a surgeon’s level of expertise and were refined with input from expert general and colorectal surgeons. One study author tested these prompts using the paid version of ChatGPT-4.5 on 18 April 2025. The paid subscription costs USD 20 per month at the time of testing.

The author iteratively evaluated responses during testing, refining prompts to ensure precise recommendations rather than generic information on colorectal cancer screening and surveillance. An additional objective was to mitigate the likelihood of ChatGPT providing no recommendation due to medicolegal disclaimers. This process involved testing prompts across all key questions until ChatGPT consistently produced relevant recommendations. The prompt engineering phase lasted approximately 12 h due to limitations in using ChatGPT. No follow-up prompts were required post-engineering. Specific societies or organizations were not referenced within the prompts to minimize bias. In total, 54 standardized prompts were developed, each reflecting distinct patient scenarios and key clinical questions (Figure 1). Prompts were initially constructed in English and subsequently translated into Romanian and Italian.

### 2.4. Query Strategy

ChatGPT-4.5 was formally queried on 18 April 2025 by four study authors. These co-investigators conducted the queries in Hamilton, Ontario, Canada; London, United Kingdom; Verona, Italy; and Cluj-Napoca, Romania. Each investigator completed all queries within 24 h in their respective location. To prevent response bias, a new chat session was initiated for each hypothetical patient case, ensuring that prior responses did not influence subsequent outputs. Additionally, the study team conducted the same queries in the native language of each country. For example, the Romanian co-author queried ChatGPT using all standardized prompts in both English and Romanian.

### 2.5. Performance Evaluation

ChatGPT’s accuracy was assessed by comparing its responses against established clinical practice guidelines. A detailed list of the guidelines and references used for each country is provided in Appendix A to support transparency and reproducibility [19,20,21]. Since comprehensive national guidelines covering all case variations were unavailable in Italy and Romania, specifically, two board-certified colorectal surgeons from each country independently reviewed the responses using a standardized scoring rubric based on regionally accepted practices. Discrepancies were resolved through consensus discussions. Expert surgeons were consulted to determine whether reactions aligned with standard practice.

A standardized data collection form was used to compile response data. One study team member conducted data analysis using descriptive statistics, reporting counts, and percentages to evaluate the alignment of ChatGPT-generated recommendations with official guidelines or expert surgeon opinions for each patient case and key question. Responses were classified as follows:Correct: Fully aligned with country-specific clinical guidelines.Partially Correct: Aligned in part but omitted key details or contained minor inaccuracies.Incorrect: Misaligned with guidelines or provided misleading recommendations.

Responses were considered non-aligned if they contradicted guidelines or failed to provide meaningful answers. Comparisons were made against country-specific guidelines; for example, responses to inquiries from Canada were evaluated against North American guidelines. Although no formal reporting checklists exist for chatbot assessments, the Chatbot Assessment Reporting Tool is currently being developed [22].

## 3. Results

A procedural diagram is provided in Figure 2.

Table 1 presents the alignment of ChatGPT-4.5-generated recommendations for colorectal cancer screening, categorized by country and case. ChatGPT recommendations were fully congruent with expert surgeon opinions or regional clinical guidelines in 27/54 (50%) cases in Canada, 34/54 (63.0%) cases in Italy, 22/54 (40.7%) cases in Romania, and 30/54 (55.6%) cases in the UK. Partial congruence was observed in 23/54 (42.6%) cases in Canada, 15/54 (27.7%) cases in Italy, 32/54 (59.3%) cases in Romania, and 17/54 (31.5%) cases in the UK.

Table 2 presents the performance of ChatGPT-4.5 across various countries, incorporating key qualitative insights from expert validation.

Table 3 compares ChatGPT-4.5’s alignment with colorectal cancer screening guidelines in English, Italian, and Romanian. ChatGPT-generated recommendations were fully congruent in 35/54 (64.8%) cases in Italian and 22/54 (40.7%) cases in Romanian. Partial congruence was observed in 14/54 (25.9%) cases in Italian and 32/54 (59.3%) cases in Romanian. In Romania, performance was consistent between English and Romanian queries, while in Italy, ChatGPT-4.5 performed slightly better in Italian than in English, except in case 51.

## 4. Discussion

The ability of ChatGPT-4.5 to provide geographically tailored recommendations for colorectal cancer screening and surveillance varied by region. Italy exhibited the highest level of alignment with conventional practices, while Romania showed the lowest. In contrast to its performance in response to prompts posed in English, ChatGPT-4.5 performed equally well in Romanian and Italian. Clinicians, researchers, and patients should recognize the limitations of using LLMs like ChatGPT for clinical advice based on their geographical location.

This study’s mixed-methods design allowed for a thorough evaluation of ChatGPT-4.5’s accuracy in colorectal cancer screening recommendations. Combining quantitative correctness rates with qualitative expert assessments offered deeper insights into geographic inconsistencies. A monomethod approach that relies solely on numerical correctness may overlook critical contextual factors, such as regional guideline differences and language nuances. By utilizing both accuracy data and expert interpretation, this study underscores the significance of methodological triangulation in evaluating AI-driven clinical decision tools.

Previous studies conducted in different geographical locations yielded varying results on the ability of ChatGPT-3.5 to answer colorectal cancer screening questions [9,10,11]. Our findings suggest that the newer ChatGPT-4.5 version performs inconsistently for surgeons across different countries, depending on local clinical practice guidelines. The development of high-quality guideline recommendations is rooted in the premise that recommendations should be formed based on evidence rather than expert opinion [23]. Multidisciplinary panels establish guidelines to review the quality and certainty of available evidence transparently and systematically [23,24,25].

This rigorous methodology produces guideline recommendations of the highest evidence quality. Therefore, the success of LLMs in generating clinical advice can be evaluated by their use of high-quality guideline recommendations. Other factors, such as stakeholder values and preferences, resource use, equity, acceptability, and implementation feasibility, may differ by region and can influence the strength and direction of guideline recommendations [25,26]. As a result, guideline recommendations sometimes vary across different regions. Our study reveals that ChatGPT-4.5 exhibits distinct performance characteristics for surgeons in various countries. This variation may be due to how LLMs like ChatGPT function. During its training, ChatGPT learns word associations by analyzing large datasets and then uses this knowledge to predict the most likely next word in its responses [5,27,28,29]. Consequently, its responses heavily depend on its training dataset and may not reflect local practices. For example, in a high-risk case involving a 38-year-old with two first-degree relatives diagnosed before age 50, ChatGPT correctly recommended starting colonoscopy at age 40 or 10 years before the earliest diagnosis. In contrast, in a post-polypectomy case involving a 65-year-old, the model wrongly advised annual screening despite guidelines recommending a 3-year surveillance interval.

LLMs that support multiple languages can generally be classified into two categories: pre-trained multilingual models and translation-based models [30,31]. Pre-trained multilingual models, such as ChatGPT, are trained to process responses in multiple languages. Conversely, translation-based models are only trained in one language and instead rely on translating information back and forth for processing and output [31]. Despite being a multilingual model, ChatGPT was primarily trained on English datasets [5,13]. Moreover, the majority of its users interact with ChatGPT in English, contributing more English data for future models to learn from. Our study found that, in the context of colorectal cancer screening, ChatGPT-4.5 performed equally well in Italian and Romanian, compared to English. The ability of ChatGPT-4.5 to perform well in multiple languages, despite limited exposure to data in other languages during pre-training, may be attributed to zero-shot learning. Zero-shot learning enables LLMs to recognize and answer questions about material they have not encountered during training. LLMs achieve this by using semantic relationships learned during pre-training to connect known and unknown variables [29,32]. Our findings are consistent with those of Grimm et al., who found that ChatGPT-4 could accurately respond to English prompts with outputs in Spanish and Mandarin [12]. Several studies have evaluated ChatGPT’s ability to answer standardized knowledge-based questions such as those on the US or Japanese Medical Licensing Exams [33,34,35]. Takagi et al. also found that ChatGPT-4 could pass the Japanese medical licensing exams [33]. Given its limited pre-training exposure to other languages, ChatGPT’s consistent performance in Italian and Romanian, without relying on translations into English for processing, demonstrates its immense potential for global use.

On average, physicians have 0.16 to 1.27 questions that they need to look up per patient encounter [36]. Despite this, only 30–57% of knowledge gaps lead to knowledge-seeking behavior [37]. The largest barriers to this are time and ineffective search skills [37,38]. LLMs can eliminate these barriers, generating responses in seconds and providing physicians with information more efficiently. As LLMs become increasingly available in other countries and languages, physicians worldwide should also be prepared to educate patients on the benefits and drawbacks of using ChatGPT for medical advice. Moreover, with healthcare shortages, hospital systems have increasingly relied on technology to increase diagnostic efficiency and minimize errors. As such, hospital managers may consider integrating LLMs into healthcare systems shortly. Given ChatGPT’s inconsistent performance in different regions, hospitals may consider developing their own private, closed models based on local institutional data to ensure LLMs align with regional standards.

A major limitation of this study is the lack of transparency regarding ChatGPT’s training sources. While OpenAI has disclosed that its model is trained on publicly available text, the proportion of the medical literature compared to general web-based content remains unknown. This opacity introduces potential biases in the chatbot’s responses, as it may favor widely discussed or frequently cited sources rather than the most clinically rigorous ones. Moreover, the dataset is primarily English-based, raising concerns about whether recommendations generated in non-English languages stem from direct multilingual training or machine translation processes.

Another significant limitation is the difference between the free and paid versions of ChatGPT. ChatGPT-4.5, accessible via the paid version, may demonstrate superior accuracy compared to its free counterpart (ChatGPT-4o), yet the specific differences in medical query processing between the two remain unclear. This poses a challenge for clinicians and researchers attempting to assess LLM performance, as studies evaluating different versions may not be directly comparable. Future investigations should explore how model versioning affects the quality of clinical recommendations and whether premium versions offer meaningful improvements in medical accuracy.

One of the most striking findings of this study is the lack of concordance between ChatGPT-generated titles and its actual recommendations. In multiple cases, the chatbot generated misleading or overly confident headings that did not align with the content of its responses. This inconsistency could lead to misinterpretation by clinicians or patients, potentially resulting in inappropriate clinical decisions. Future work should examine the extent of this issue and whether prompt engineering strategies can help mitigate it.

Lastly, our study highlights the absence of unified colorectal cancer guidelines across different countries, particularly in Romania and Italy, where gaps in surveillance protocols were observed. ChatGPT’s varied accuracy in these locations reflects not only the limitations of the model itself but also the lack of comprehensive national policies governing CRC screening and post-treatment surveillance. Policymakers should consider strengthening country-specific guidelines to reduce discrepancies in AI-driven recommendations and enhance uniformity in colorectal cancer care.

Currently, the use of ChatGPT in healthcare is mostly experimental and lacks standardized frameworks for integration into clinical decision-making. Future efforts should concentrate on creating uniform protocols for prompt design, response validation, and ongoing model calibration. Standardization needs to include institution-specific fine-tuning of models to align with local guidelines, terminology, and population health traits, as well as version control and transparency regarding training data and model limits. Additionally, integrating AI into electronic health records (EHRs) to provide real-time, guideline-concordant recommendations and establishing governance models with multidisciplinary oversight and regular audits of outputs are essential.

## 5. Conclusions

In conclusion, this study evaluated the geographic and linguistic accuracy of ChatGPT-4.5 in providing colorectal cancer screening and surveillance recommendations. Although the model showed potential, its performance varied significantly across countries and languages, highlighting the need for contextual validation. Future versions of LLMs must incorporate high-quality, region-specific clinical data to ensure safe use in real-world settings. Additionally, its limitations in data sourcing, temporal relevance, version differences, and title–recommendation misalignment require cautious interpretation. Healthcare professionals must stay vigilant when using AI-generated advice in clinical practice and advocate for greater transparency and standardization in LLM development. 

## Figures and Tables

**Figure 1 jcm-14-05101-f001:**
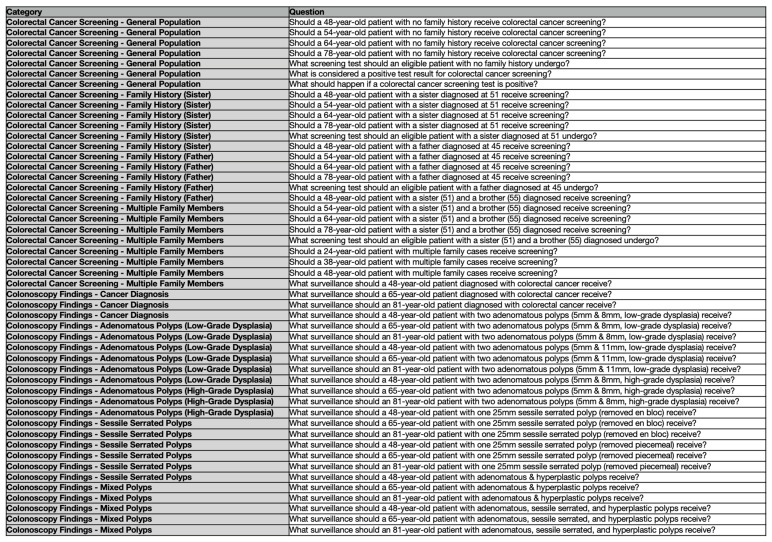
Key clinical questions submitted to ChatGPT.

**Figure 2 jcm-14-05101-f002:**
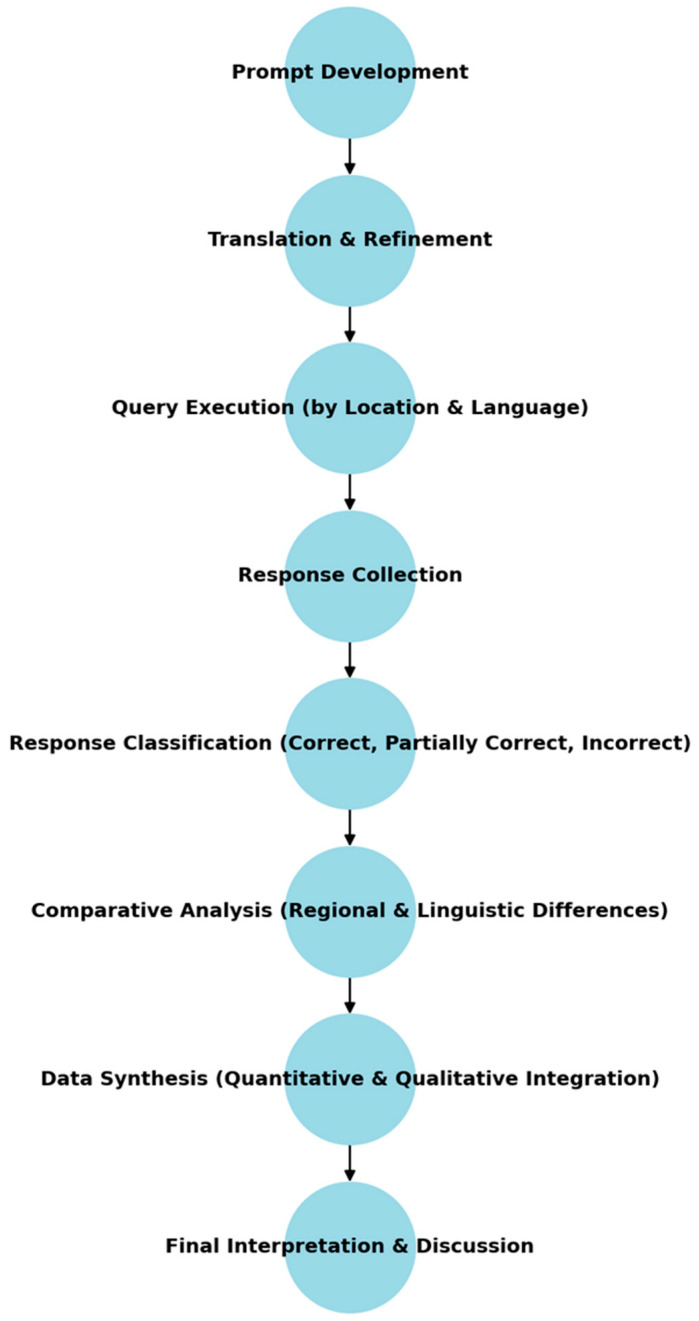
Procedural diagram of the study.

**Table 1 jcm-14-05101-t001:** Alignment of ChatGPT-4.5-generated recommendations with regional guidelines.

Case	Canada	Italy	Romania	UK
Frequency Completely Aligned with Recommendations	27/54	34/54	22/54	30/54
Frequency Partially Aligned with Recommendations	23/54	15/54	32/54	17/54
1	No	Yes	Partial	Yes
2	Yes	Yes	Yes	Yes
3	Yes	Yes	Yes	Yes
4	Yes	Yes	Yes	Yes
5	Partial	Partial	Yes	Yes
6	Partial	Yes	Yes	Partial
7	Partial	Yes	Yes	Yes
8	Partial	Partial	Partial	No
9	Partial	Partial	Partial	No
10	Partial	Partial	Partial	Yes
11	Partial	Yes	Yes	Partial
12	Partial	Yes	Yes	Partial
13	Partial	Partial	Partial	Partial
14	Partial	Partial	Partial	Partial
15	Partial	Partial	Partial	Partial
16	Yes	Partial	Yes	Yes
17	Partial	Yes	Yes	Partial
18	No	Partial	Yes	Partial
19	Yes	Partial	Partial	Yes
20	Yes	Partial	Yes	Yes
21	Yes	Partial	Yes	Yes
22	Yes	Yes	Yes	Yes
23	Yes	Yes	Yes	Partial
24	Partial	Yes	Yes	Partial
25	Partial	Yes	Yes	Yes
26	Partial	Yes	Yes	Yes
27	Partial	Yes	Yes	Yes
28	Partial	Yes	Yes	Yes
29	Yes	Yes	Yes	Partial
30	Yes	Yes	Yes	Partial
31	Yes	Yes	Partial	Partial
32	Partial	Yes	Partial	Partial
33	Partial	Yes	Partial	Partial
34	No	Yes	Partial	No
35	No	Yes	Partial	Partial
36	Partial	No	Partial	Yes
37	Yes	No	Partial	Yes
38	Yes	Yes	Partial	Yes
39	Yes	No	Partial	Yes
40	Yes	No	Partial	Yes
41	Yes	Yes	Partial	Yes
42	Yes	Yes	Partial	Yes
43	Yes	Yes	Partial	Yes
44	Yes	Yes	Partial	Yes
45	Yes	Yes	Partial	No
46	Yes	Yes	Partial	No
47	Yes	Yes	Partial	Yes
48	Yes	Yes	Partial	Partial
49	Yes	Yes	Partial	No
50	Yes	Yes	Partial	Yes
51	Yes	Partial	Partial	Yes
52	Partial	Partial	Partial	Yes
53	Partial	Partial	Partial	Yes
54	Partial	No	Partial	No

**Table 2 jcm-14-05101-t002:** Alignment of numerical performance rates with key qualitative insights from expert validation.

Country	Correct (%)	Partially Correct (%)	Incorrect (%)	Commentary
Canada	50%	42.6%	7.4%	Guidelines well-defined; minor inconsistencies observed
Italy	63%	27.7%	9.3%	More flexible guidelines; chatbot struggles with ambiguity
Romania	40.7%	59.3%	0%	Lack of formal guidelines; expert judgment required
UK	55.6%	31.5%	12.9%	Strong national guidelines, but chatbot misinterpreted surveillance intervals

**Table 3 jcm-14-05101-t003:** Alignment of ChatGPT-4.5-generated recommendations in various languages.

Case	English (Canada)	Italian	Romanian	English (UK)
Frequency Completely Aligned with Recommendations	27/54	35/54	22/54	30/54
Frequency Partially Aligned with Recommendations	23/54	14/54	32/54	17/54
1	No	Yes	Partial	Yes
2	Yes	Yes	Yes	Yes
3	Yes	Yes	Yes	Yes
4	Yes	Yes	Yes	Yes
5	Partial	Partial	Yes	Yes
6	Partial	Yes	Yes	Partial
7	Partial	Yes	Yes	Yes
8	Partial	Partial	Partial	No
9	Partial	Partial	Partial	No
10	Partial	Partial	Partial	Yes
11	Partial	Yes	Yes	Partial
12	Partial	Yes	Yes	Partial
13	Partial	Partial	Partial	Partial
14	Partial	Partial	Partial	Partial
15	Partial	Partial	Partial	Partial
16	Yes	Partial	Yes	Yes
17	Partial	Yes	Yes	Partial
18	No	Partial	Yes	Partial
19	Yes	Partial	Partial	Yes
20	Yes	Partial	Yes	Yes
21	Yes	Partial	Yes	Yes
22	Yes	Yes	Yes	Yes
23	Yes	Yes	Yes	Partial
24	Partial	Yes	Yes	Partial
25	Partial	Yes	Yes	Yes
26	Partial	Yes	Yes	Yes
27	Partial	Yes	Yes	Yes
28	Partial	Yes	Yes	Yes
29	Yes	Yes	Yes	Partial
30	Yes	Yes	Yes	Partial
31	Yes	Yes	Partial	Partial
32	Partial	Yes	Partial	Partial
33	Partial	Yes	Partial	Partial
34	No	Yes	Partial	No
35	No	Yes	Partial	Partial
36	Partial	No	Partial	Yes
37	Yes	No	Partial	Yes
38	Yes	Yes	Partial	Yes
39	Yes	No	Partial	Yes
40	Yes	No	Partial	Yes
41	Yes	Yes	Partial	Yes
42	Yes	Yes	Partial	Yes
43	Yes	Yes	Partial	Yes
44	Yes	Yes	Partial	Yes
45	Yes	Yes	Partial	No
46	Yes	Yes	Partial	No
47	Yes	Yes	Partial	Yes
48	Yes	Yes	Partial	Partial
49	Yes	Yes	Partial	No
50	Yes	Yes	Partial	Yes
51	Yes	Yes	Partial	Yes
52	Partial	Partial	Partial	Yes
53	Partial	Partial	Partial	Yes
54	Partial	No	Partial	No

## Data Availability

The original contributions presented in this study are included in the article/Appendix A. Further inquiries can be directed to the corresponding author.

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

cancer screening. Epidemiol Prev..

