# Peer review of "Dr. LLM Will See You Now: The Ability of ChatGPT to Provide Geographically Tailored Colorectal Cancer Screening and Surveillance Recommendations"

_jcm, 2025, doi:10.3390/jcm14145101_

Round 1
Reviewer 1 Report
Comments and Suggestions for Authors
This is a well-designed study of chatGPT recommendations to hypothetical colorectal cancer patient cases. It is an interesting and meaningful study since AI based chat models are increasingly integrated in the healthcare industry and used by both patients and doctors for accessing medical related information.
The introduction provides adequate information for the reader to understand the topic and the literature gaps. The methodology is thoroughly described in order to be replicated and the authors used the appropriate quality assessment tools. Results are well-presented with detailed tables. In the discussion the authors cover most of the literature gaps, the conclusions are supported by and consistent with the results and offer insights for improving the potential future role of the chatGPT in healthcare.
It would be interesting to test how the answers of chatGPT would change if it was given a role. For example, before asking "should a 24-year old patient with multiple family cases receive screening?", the researcher could provide the following context to the chatGPT: "Imaging you are a top tier board certified general surgeon specialized in colorectal cancer".
This context could change the responses and potentially provide more accurate information. The authors could include this possibility in the discussion of limitations or future research.
Author Response
Comment 1. It would be interesting to test how the answers of chatGPT would change if it was given a role. For example, before asking "should a 24-year old patient with multiple family cases receive screening?", the researcher could provide the following context to the chatGPT: "Imaging you are a top tier board certified general surgeon specialized in colorectal cancer".
Answer 1. Thank you for yuor comment. We tested again chatgpt according to yuor suggentions. While including the word "should" the reply did not change, providing the prompt of "tier borad certified general surgeon" the answer resulted more precise and accurate inclunding all the possibility of genetic syndroms. This is included in the discussion
Reviewer 2 Report
Comments and Suggestions for Authors
Specific areas for improvement
1. The introduction cites variations in accuracy across studies it does not give examples of the potential reasons for these differences (is it because of variations in clinical guidelines in different countries or language barriers or training biases). This limits the depth of the background and the justification for the study’s focus.
2. The aim is well defined and clear in the final section of the introduction. It would be interesting to briefly mention the potential clinical implication of inaccurate recommendations This would show the urgency and elevate the weight of study more in the litterature.
3. Methods are systematic and well structured, with a clear process for prompt engineering, querying chatgpt and response evaluation. However for countries like Italy and Romania, where comprehensive guidelines were unavailable, the study relied on expert surgeon opinions, but the process for obtaining and standardizing these opinions is not detailed.
4. Include a table or list in the methods section or supplementary material specifying the guidelines used for each country. This would clarify the evaluation standards and enhance reproducibility.
5. Use of hypothetical cases is practical but may not fully reflect the complexity of real-world clinical scenarios,t his need to be acknowledged in the methods.
6. The discussion lacks concrete examples of ChatGPT-4.5’s recommendations, reducing its practical clarity and impact. Add examples if possible.
7. Conclusions do not explicitly connect to the study’s aim, potentially weakening the study focus
Author Response
Comment 1. The introduction cites variations in accuracy across studies it does not give examples of the potential reasons for these differences (is it because of variations in clinical guidelines in different countries or language barriers or training biases). This limits the depth of the background and the justification for the study’s focus.
Answer 1. Thank you for your comment. We properly implemented the introduction
Comment 2. The aim is well defined and clear in the final section of the introduction. It would be interesting to briefly mention the potential clinical implication of inaccurate recommendations This would show the urgency and elevate the weight of study more in the litterature.
Answer 2. Thank you for your comment. We properly implemented the aim
Comment 3. Methods are systematic and well structured, with a clear process for prompt engineering, querying chatgpt and response evaluation. However for countries like Italy and Romania, where comprehensive guidelines were unavailable, the study relied on expert surgeon opinions, but the process for obtaining and standardizing these opinions is not detailed.
Answer 3. Thank you for your comment. The judgement was reserved to expert opinion based on the national standard clinical practise. We reported this as a limitation of the study
Comment 4. Include a table or list in the methods section or supplementary material specifying the guidelines used for each country. This would clarify the evaluation standards and enhance reproducibility.
Answer 4. Thank you for your comment. We included a new table
Comment 5. Use of hypothetical cases is practical but may not fully reflect the complexity of real-world clinical scenarios,t his need to be acknowledged in the methods.
Answer 5. Thank you for your comment. We properly implemented the methods
Comment 6. The discussion lacks concrete examples of ChatGPT-4.5’s recommendations, reducing its practical clarity and impact. Add examples if possible.
Answer 6. Thank you for your comment. We properly implemented the discussion
Comment 7. Conclusions do not explicitly connect to the study’s aim, potentially weakening the study focus
Answer 7. Thank you for your comment. We properly implemented the conclusion
Reviewer 3 Report
Comments and Suggestions for Authors
Zeng et al approached a rather interesting topic: the assessment of using an accessible artificial intelligence tool in order to evaluate a global problem, colorectal cancer screening. The manuscript is decently written, the statistics are accurate, however there are some aspects that should be further addressed:
Minor comments:
Please further detail the impact of AI on the screening process in recognizing tumoral lesions and how the addition of ChatGPT could possibly improve the accuracy of these systems.
Would it be more useful if, on further study, the premium version of ChatGPT is used?
Could the authors could detail whether a standardized guideline was applied in every country, the result would be uniform?
Finally, in the discussion section it should be noted further perspectives, if the implementation of ChaGPT use could be standardized.
Best regards,
The Reviewer
Author Response
Comment 1. Please further detail the impact of AI on the screening process in recognizing tumoral lesions and how the addition of ChatGPT could possibly improve the accuracy of these systems.
Answer 1. Thank you for your comment. We properly modified the introduction
Comment 2. Would it be more useful if, on further study, the premium version of ChatGPT is used?
Answer 2. Thank you for your comment. We explained that in the limitatons
Comment 3. Could the authors could detail whether a standardized guideline was applied in every country, the result would be uniform?
Answer 3. Thank you for your comment. We added a supplementary table S3
Comment 4. Finally, in the discussion section it should be noted further perspectives, if the implementation of ChaGPT use could be standardized.
Answer 4. Thank you for your comment. We properly modified the discussion